# A Fully Automated Analysis Pipeline for 4D Flow MRI in the Aorta

**DOI:** 10.3390/bioengineering12080807

**Published:** 2025-07-27

**Authors:** Ethan M. I. Johnson, Haben Berhane, Elizabeth Weiss, Kelly Jarvis, Aparna Sodhi, Kai Yang, Joshua D. Robinson, Cynthia K. Rigsby, Bradley D. Allen, Michael Markl

**Affiliations:** 1Department of Radiology, Northwestern University, Chicago, IL 60611, USAcrigsby@luriechildrens.org (C.K.R.);; 2Department of Medical Imaging, Lurie Children’s Hospital, Chicago, IL 60611, USA; 3Department of Cardiology, Lurie Children’s Hospital, Chicago, IL 60611, USA

**Keywords:** 4D flow MRI, automated analysis, AI-driven automation

## Abstract

Four-dimensional (4D) flow MRI has shown promise for the assessment of aortic hemodynamics. However, data analysis traditionally requires manual and time-consuming human input at several stages. This limits reproducibility and affects analysis workflows, such that large-cohort 4D flow studies are lacking. Here, a fully automated artificial intelligence (AI) 4D flow analysis pipeline was developed and evaluated in a cohort of over 350 subjects. The 4D flow MRI analysis pipeline integrated a series of previously developed and validated deep learning networks, which replaced traditionally manual processing tasks (background-phase correction, noise masking, velocity anti-aliasing, aorta 3D segmentation). Hemodynamic parameters (global aortic pulse wave velocity (PWV), peak velocity, flow energetics) were automatically quantified. The pipeline was evaluated in a heterogeneous single-center cohort of 379 subjects (age = 43.5 ± 18.6 years, 118 female) who underwent 4D flow MRI of the thoracic aorta (*n* = 147 healthy controls, *n* = 147 patients with a bicuspid aortic valve [BAV], *n* = 10 with mechanical valve prostheses, *n* = 75 pediatric patients with hereditary aortic disease). Pipeline performance with BAV and control data was evaluated by comparing to manual analysis performed by two human observers. A fully automated 4D flow pipeline analysis was successfully performed in 365 of 379 patients (96%). Pipeline-based quantification of aortic hemodynamics was closely correlated with manual analysis results (peak velocity: *r* = 1.00, *p* < 0.001; PWV: *r* = 0.99, *p* < 0.001; flow energetics: *r* = 0.99, *p* < 0.001; overall *r* ≥ 0.99, *p* < 0.001). Bland–Altman analysis showed close agreement for all hemodynamic parameters (bias 1–3%, limits of agreement 6–22%). Notably, limits of agreement between different human observers’ quantifications were moderate (4–20%). In addition, the pipeline 4D flow analysis closely reproduced hemodynamic differences between age-matched adult BAV patients and controls (median peak velocity: 1.74 m/s [automated] or 1.76 m/s [manual] BAV vs. 1.31 [auto.] vs. 1.29 [manu.] controls, *p* < 0.005; PWV: 6.4–6.6 m/s all groups, any processing [no significant differences]; kinetic energy: 4.9 μJ [auto.] or 5.0 μJ [manu.] BAV vs. 3.1 μJ [both] control, *p* < 0.005). This study presents a framework for the complete automation of quantitative 4D flow MRI data processing with a failure rate of less than 5%, offering improved measurement reliability in quantitative 4D flow MRI. Future studies are warranted to reduced failure rates and evaluate pipeline performance across multiple centers.

## 1. Introduction

Time-resolved 3D phase contrast (4D flow) MRI is a flexible technique that can be used to measure and quantify cardiovascular hemodynamics. For example, 4D flow MRI of the thoracic aorta can assess peak velocities (*V*_max_) in the ascending aorta [1,2,3], a critical metric for the management of aortic valve diseases [4], or to estimate global aortic pulse wave velocity (PWV) of the aorta, a measure of aortic stiffness that has been shown to be associated with the development of aortopathy [5,6]. However, 4D flow MRI data processing often requires manual and cumbersome human input at several stages, from setting a pre-processing threshold for phase correction parameters [7] to manual 3D aortic segmentation of vessels and/or regions of interest. This poses a significant challenge for conducting studies with large cohorts, as the resultant workload to process data (up to 10–20 min. per dataset for experienced users) can become untenable. For example, the processing sub-step of creating an accurate segmentation for the aortic vessel typically requires over ten minutes of active attention, even from an experienced user [8,9]. End-to-end manual analysis of 4D flow data using commercial software is reported to consume over 20 min per dataset [10]. Such software includes automation steps, such as offering a preliminary delineation of aortic geometry, but still requires manual user input at several points to set pre-processing correction thresholds and refine segmentations [10]. In addition, the need for manual user input can introduce observer bias and limit reproducibility [10].

Recent advances in artificial intelligence (AI)-based processing techniques applied to 4D flow MRI have created an opportunity to dramatically reduce the required amount of manual processing for such datasets [8,11,12,13,14,15]. Several recent works have presented deep learning networks that can replace manual processing for 4D flow analysis tasks, such as identifying regions of static tissue, performing background phase offset correction and noise masking, correcting velocity aliasing, and creating 3D segmentations of the aortic regions [8,11,12,13,14,15,16,17]. However, the scope of prior work has been limited to establishing performance of AI methods for individual processing steps; a comprehensive evaluation of end-to-end automated 4D flow processing has not been shown. Given that processing tasks required for 4D flow datasets have a uniform structure, these tools can be incorporated into a pipeline framework to facilitate high-throughput, automated processing of 4D flow MRI, and an evaluation of completely automated processing can be undertaken.

Here, we present a framework that allows the fully automated quantification of key hemodynamic metrics of the thoracic aorta, such as peak systolic velocity (*V*_max_) and pulse wave velocity (PWV), and metrics of blood flow energetics, such as viscous energy loss (EL) and kinetic energy (KE), from 4D flow MRI data. The framework was designed to include robust fail-safes for the automated processing steps to enable verifying the integrity of results, and to identify any processing failures in AI-driven tasks for subsequent correction. Our goal was to apply an AI analysis pipeline for automated 4D flow processing across a large and heterogeneous study cohort to test the following hypotheses: 1. The quantification of hemodynamic parameters using fully automated processing has close agreement with manual 4D flow MRI analysis (reference standard). 2. Manual processing creates variability in hemodynamic quantifications that can be minimized by fully automating all processing steps.

## 2. Methods

### 2.1. Analysis Pipeline Architecture for 4D Flow MRI

An outline of the processing steps for 4D flow MRI data analysis was developed and used as a framework for the analysis pipeline (Figure 1). This framework was implemented as a script taking input of a spreadsheet of 4D flow patient data to be processed (Figure 2), executing a series of user-defined processing steps. The pipeline was designed to accept flexible input data, including DICOM images from a 4D flow MRI scan or previously processed 4D flow data (magnitude and velocity data with aortic segmentation). If DICOM images were listed as input data in the spreadsheet, any user-specified pre-processing corrections were applied, and an aortic 3D segmentation was generated prior to performing hemodynamic quantifications. Alternatively, if previously processed 4D flow data were specified as the input, only the processing steps to complete hemodynamic quantifications were invoked. 

This pipeline design was chosen to simplify user interaction by coalescing all details of the data processing into a single input data spreadsheet. The computational flow for the pipeline (Figure 1) was structured to iterate through all subject datasets specified in the data spreadsheet and perform four basic tasks: [A] automatically determine what processing is required given the input data; [B] complete all required preprocessing tasks using traditional or AI-based methods [11,17]; [C] invoke AI-based aortic 3D segmentation [8,16] as needed; and [D] complete all hemodynamic parameter quantification steps specified in the spreadsheet by the user (Figure 2A).

### 2.2. Analysis Pipeline for 4D Flow MRI—Processing Tasks

The modules for performing pre-processing tasks included threshold-based noise masking (data list keyword: “noise”), background phase correction (“phase”), and fixed-iteration phase-aliasing unwrapping (“alias”), each with pre-defined, fixed thresholds [7]. Additionally, modules were implemented for performing AI-based operation of these steps without any manually defined parameters (“noiseai”, “phaseai”, and “aliasai”) [11,17]. The traditional noise, phase, and alias modules utilized a supplemental input at the beginning of processing that specified the threshold values to be used. The noiseai, phaseai, and aliasai modules required no user input to perform corrections, which was accomplished by applying previously trained neural networks to identify the image regions to be used as a basis for performing the applicable corrections [11,17]. The networks supporting the pre-processing modules were trained on 4D flow MRI data from both BAV and TAV subjects with aortopathy, in addition to data from healthy control subjects [8,11]. Both adult and pediatric subjects were represented in the training data; the cohort was approximately 80% adult [8]. Next, previously trained deep learning 3D U-nets [8,16] were used to create a segmentation mask for the entire aorta and three subregions: ascending aorta (AAo), arch, and descending aorta (DAo). The networks for performing aortic segmentations were trained with the same subject datasets, as were the pre-processing networks [8]. The data used for training are not publicly available due to protected health information regulations and institutional policies.

The 3D aorta segmentations were used to mask the 4D flow velocity data, followed by the quantification of several hemodynamic parameters: regional (AAo, arch, DAo) systolic peak velocity (*V*_max_), kinetic energy (KE), and viscous energy loss (EL), as well as global aortic pulse wave velocity (PWV). The computational methods for performing these quantifications are based on previously described techniques and algorithms [1,18,19,20,21]. Briefly, systolic peak velocity was calculated as the maximum velocity in the region of interest after applying an outlier removal filter (sort values and remove those exceeding ten times the mean slope of the top 1%) [1,19]; kinetic energy was calculated as KE = 0.5 *ρ dV v*^−2^, where blood density ρ has a value of 1.06 g/cm^3^, and *dV* is the acquired voxel volume [21]; viscous EL was quantified from μ *Φ dV*, where the dynamic viscosity of blood μ is 3.2 cP and *Φ* is the viscous portion of the incompressible Navier–Stokes equation [18]; and pulse wave velocity was determined by the cross-correlation of flow–time curves along the entire aortic length, as described previously [20].

### 2.3. Analysis Pipeline for 4D Flow MRI—Quality Control 

Any pipeline execution errors were logged via text output. To enable a rapid review of the processing results’ quality for each dataset, a quality check module generated standardized visualizations of AI-generated 3D aorta segmentations, including an aortic centerline with test planes spaced 25mm apart through the aortic length for calculating through-plane flow curves (Figure 3). In addition, overlays of magnitude, velocity images, and aortic segmentation were generated (Figure 3). These quality check images enable the visual assessment of segmentation quality and potential errors in the quantification of velocity data. An absence of branching vessels from the aorta (e.g., brachiocephalic artery, carotid artery, etc.) was not considered an error. For evaluating the overall success in the dataset processing, all quality check outputs were systematically reviewed. The quality was considered to be acceptable if a contiguous region including all of the thoracic ascending, arch, and descending aorta was included, and if no extraneous regions—such as the pulmonary artery or other thoracic anatomy—were included (Figure 3).

### 2.4. Technical Implementation of Processing Pipeline

The processing pipeline was structured to parse the input data spreadsheet and identify all processing steps required for each dataset, then to build arrays of function handles referencing the modules to be invoked for each dataset (Figure 2B), and finally to apply all appropriate processing modules to each dataset. The processing modules were implemented as encapsulated routines using a uniform input structure that included a file reference to the magnitude and velocity data, the vessel segmentation, and a file location to write output. This design facilitated generic handling of the processing modules to ensure all modules defined in the input spreadsheet could be invoked using the same input and output coding (Figure 2). The input spreadsheet format allowed for the mixed processing of data, for example, calculating PWV in one dataset with an existing 3D segmentation, while completing DICOM file conversion/pre-processing/segmentation/hemodynamic quantification in the following dataset (Figure 2).

A standardized data structure was used for storing 4D flow magnitude and velocity data in a compact, compressed (HDF5 standard-based) format rather than single-slice and single-time-point DICOM images with duplicative metadata. The data structure included a header listing essential scan parameters (voxel size, temporal resolution, physical volume coordinates, VENC).

### 2.5. Study Cohort and Image Acquisition

Data from a heterogeneous cohort of subjects with 4D flow MRI was assembled to evaluate pipeline performance (Table 1 and Table 2). We retrospectively included patients who underwent 4D flow MRI data as part of a standard-of-care cardiac MRI exam at Northwestern University (NU) and Lurie Children’s Hospital (LCH). An NU database for bicuspid aortic valve (BAV) patients with clinical cardiothoracic MR imaging scans from 2012 to 2019 was queried and filtered to include only subjects with complete 4D flow image datasets acquired in sagittal oblique orientation that were available for analysis. At LCH, an imaging database of patients who received 4D flow MRI between 2012 and 2022 was queried to include patients with the following connective tissue disorders: Marfan syndrome, Loeys–Dietz syndrome, and Ehler–Danlos syndrome. Further, the study cohort included a group of patients with aortic valve prostheses (aortic valve replacement with mechanical prosthesis by manufacturers On-X [Kennesaw, GA, USA], St. Jude [Little Canada, MN, USA], or Carbomedics [Austin, TX, USA]). Finally, healthy volunteer data were prospectively collected as part of a research cardiac MRI scan. All subjects were enrolled after approval from the NU and Lurie Children’s institutional review boards (IRBs), with a waiver of consent for retrospective inclusion or informed consent for prospective recruitment.

All imaging data were acquired using a 4D flow MRI sequence at 1.5 T or 3 T, with prospective or retrospective cardiac gating, respiratory navigation, with spatial/temporal resolution 1.5–2.5 × 1.5–2.5 × 1.5–4.5 mm^3^/30–42 ms, *R* = 2 (GRAPPA) or *R* = 8–10 (GRAPPA + compressed sensing) [22,23], VENC 80–300 cm/s. Acquisitions were performed with sagittal–oblique coverage of the entire thoracic aorta.

### 2.6. Study Design and Reference Standards

All 4D flow data datasets were reviewed for pipeline processing errors using the quality check outputs (Figure 3). Datasets with errors were excluded from subsequent comparisons of hemodynamic parameters. For the sub-cohorts of BAV patients and controls, data were manually processed twice by different human observers (“manual_1_”, “manual_2_”; divided among six total observers), which served as the reference standard for evaluation of pipeline performance. The individual observers were not matched for experience. Observers had between one and five years of experience working with conventional 4D flow MRI data analysis, and all had been trained by at least one other observer with prior data processing experience. In addition, the 4D flow processing pipeline was executed twice to demonstrate stability of computation. Two aspects of the time required for processing were quantified using file creation and modification timestamps: 1. user set-up duration (the time required to prepare the input spreadsheet); 2. end-to-end net computational processing time, including all hemodynamic quantifications, for twenty representative datasets (ten healthy controls, ten BAV patients) when performed using a desktop workstation (Intel i7-7700 CPU; no GPU used), with all data stored locally (SATA HDD).

### 2.7. Statistics

Comparisons were performed for automated vs. manual_1_, automated vs. manual_2_, manual_1_ vs. manual_2_, and automated vs. automated re-run (Figure 4). To evaluate whether differences in hemodynamic parameters observed across subject groups are preserved by automated pipeline processing, additional comparisons were performed between BAV patients and healthy control groups. Correlation coefficients and linear fits were computed, and Bland–Altman plots were generated to evaluate manual vs. automated analysis results. Group-wise comparisons used a one-sample or two-sample *t*-test as appropriate for the groups compared.

## 3. Results

### 3.1. Study Cohort

The complete study cohort (Table 1 and Table 2) included 4D flow data from 379 subjects and comprised 147 adult healthy controls, 147 adult patients with BAV disease, 10 adult aortic valve disease patients with mechanical valve prostheses, 62 pediatric Marfan syndrome (MFS) patients, 11 pediatric Loeys–Dietz syndrome (LDS) patients, and 2 pediatric Ehler–Danlos syndrome (EDS) patients. Distributions of age and sex varied between subject types in the overall cohort (Table 1). The sub-cohorts of BAV subjects (*n* = 147) and healthy controls (*n* = 101) were matched in age distribution but not in sex distribution.

### 3.2. Automated AI-Based Analysis Pipeline Performance—Success Rate and Processing Time

A review of the logged output and quality check visualizations resulted in the exclusion of 14 of 379 (4%) 4D flow MRI datasets which had errors arising from incomplete or erroneous AI-based 3D aorta segmentation. The net success rate was, thus, 96% in 4D flow datasets processed without any manual intervention. For the adult 4D flow dataset, the net success rate was higher, at 98%, while in the pediatric data, the success rate was lower, at 88%. The user set-up duration (the creation and population of the input data spreadsheet) was 20.9 min. The data processing duration was 36.6 ± 4.9 min per dataset in the representative sample.

### 3.3. Automated AI-Based Analysis Pipeline Performance—Hemodynanamic Quantification

Comparison of the automated pipeline analysis vs. the manual quantification of hemodynamic parameters in healthy controls and BAV subjects (Figure 5) showed a high correlation (*R* = 0.99–1.00, *p* < 0.001) for all parameters quantified (global aortic PWV, regional aortic *V*_max_, KE, and EL). Bias (mean difference) in the quantifications was low for all parameters (*V*_max_: 0.00 m/s, PWV: −0.05 m/s, KE: 0.03 μJ and EL: 0.00 μJ, respectively), and limits of agreement were also close for all parameters (*V*_max_: 0.49 m/s, PWV: 0.11 m/s, KE: 0.17 μJ and EL: 0.01 μJ, respectively).

In the manual analysis of 4D flow MRI, there was good-to-moderate agreement between the human observers, with limits of agreement for PWV, *V*_max_, KE, and EL of ±0.68 m/s, ±0.06 m/s, ±0.24 μJ, and ±0.02 μJ. Bias was low in all parameters, while agreement for KE was the most discrepant (Figure 6 and Figure 7). 

Comparison of pipeline vs. human quantification of the same parameters demonstrated similar performance (Figure 6 and Figure 7). The limits of agreement for pipeline vs. manual_1_ or manual_2_ were *V*_max_: ±0.49 m/s or ±0.60 m/s, PWV: ±0.11 m/s or ±0.09 m/s, KE: ± 0.17 μJ or ±0.25 μJ, and EL: ± 0.01 μJ or ± 0.02 μJ, with low bias for all parameters. Lastly, no differences in hemodynamic parameters were observed for repeat quantification with the 4D flow processing pipeline (all parameters: 0 bias, 0 LoA).

### 3.4. Automated AI-Based Analysis Pipeline Performance—BAV Patients vs. Controls

The metrics derived from automated or manual processing had comparable differences between healthy subjects and BAV patients (Figure 8; auto-bav vs. auto-ctrl or manu-bav vs. manu-ctrl). The 4D flow pipeline analysis detected significant (*p* < 0.005) differences between groups for *V*_max_, KE, and EL, with close agreement to the results from the manual analysis (the reference standard) — elevated *V*_max_ in BAV patients: 33% from auto-processing or 36% from manual processing; elevated KE: 61% auto or 62% manual; elevated EL: 36% auto or 38% manual. No significant differences were observed between groups for PWV (median 4.0 m/s auto-bav or 3.9 m/s manu-bav vs. 3.7 m/s auto-ctrl or 3.8 m/s manu-ctrl). 

## 4. Discussion

Herein, a fully automated 4D flow MRI analysis pipeline using AI processing steps was developed and successfully applied in a large study cohort. The main findings of our study include the following: (1) deep learning 4D flow processing tasks and algorithms for hemodynamic quantification were successfully integrated into a pipeline analysis approach to achieve a fully automated 4D flow analysis with a high success rate of 96%, (2) the pipeline analysis demonstrated good-to-excellent agreement in terms of aortic hemodynamic parameters compared to a standard manual 4D flow analysis by human observers across a large and heterogeneous study cohort, and (3) the pipeline 4D flow analysis closely reproduced hemodynamic differences between the BAV patients and controls.

The simplified input structure and flexibility of the processing specification allow for a wide range of applications of this tool, such as simply converting a large batch of DICOM images to a standardized data structure, or creating aortic segmentations and calculating a variety of hemodynamic parameters from 4D flow MRI data. The software design principles of modularity and encapsulation were used to structure processing functionality, allowing the general framework to be extended to new applications, for example, analyzing 4D flow MRI in the brain if tools for automated brain vessel segmentation and cerebral flow quantification are created. Some efforts to create such tools have been reported in recent years for carotid and intracranial vasculature [24,25,26], although the complexity and anatomical variants of such vessels pose additional challenges compared to the thoracic aorta. The software encapsulation used here allows for the straightforward incorporation of updates to any component, such as a retrained AI for improved aortic segmentation. Utilizing an input structure that permits batch processing allows for the concentration of user attention at the beginning and end of processing, and as such reduces the total amount of time required for a user to focus on data processing. In our evaluation, the net time required for preparation of the input data spreadsheet was approximately 20 min. The processing pipeline tool was then used to process 379 subject datasets, which otherwise would have required an estimated 95 h of user interaction if performed manually (15 min per dataset). When the automated processing was complete, the review of the quality check outputs was not timed, but we estimate it to have been completed in under one hour. Thus, the amount of attended time (input preparation and output review) for performing data processing was reduced from an estimated 95 h to under an estimated 1.5 h for the moderately large cohort analyzed here. While not demonstrated here, if assuming that the attended time required per subject scales linearly or sub-linearly with this processing pipeline approach, it makes the analysis of large-cohort (*n* > 1000) datasets achievable. Adequate time must also be allocated for allowing the processing to be completed, but the computational task is fully parallelizable, and thus, computation time vs. resources can be directly traded as applicable.

Our evaluation of the overall processing success with our automated approach revealed higher rates of successful processing for adult datasets than for pediatric ones. This may be a reflection of the much higher degree of representation for adult data in the training data (80% adult). In evaluating the reliability of hemodynamic quantification, the agreement from manual_1_ to manual_2_ for the parameters evaluated here was acceptable, and it would not be expected to introduce significant confounds in a large-cohort (>100 subjects) study. However, in smaller cohorts, it could pose a challenge for observing subtler hemodynamic differences, with the degree of inter-observer variability potentially overwhelming the effect size. In contrast, the automated AI-driven process had no “inter-observer” discrepancy, since the output is completely deterministic, and the exact same quantifications result from repeated application. While such a comparison of the method against itself is somewhat tautological, it illustrates how the automation approach completely eliminates inter-observer variability by removing all human input. Furthermore, the group-wise differences between healthy controls and BAV were consistent with previous studies of BAV hemodynamics that included a comparison with healthy normal values [27,28,29]. In particular, BAV is known to coincide with elevated peak velocities, but aortic stiffness assessed using PWV is not typically elevated unless a significant degree of valve dysfunction is present [27,28,29]. Concordant with such prior findings, we observed elevated velocities and energetics in BAV subjects compared to healthy subjects, but no elevation of PWV in BAV subjects [18,30,31]. Currently, the clinical management of BAV primarily considers peak aortic velocity and, otherwise, does not make extensive use of advanced hemodynamic parameters such as PWV or flow energetics [4,32], but recent research into disease pathways for BAV patients has found accruing evidence that these parameters may play an important role in the risk for aortic complications [33,34,35]. Establishing reliable hemodynamic quantification methods, as we have undertaken here, is a crucial step in establishing the scientific evidence required for translating the use of these parameters to clinical decision making.

### Study Limitations

No ground truth reference values were available to assess the accuracy of any observer (human or AI), but the AI-based quantifications did not show significant bias when compared to either set of human observations. This suggests that the automated quantifications should not be in any lower regime of accuracy than the current standard with manual quantification steps. Our human (manual) observations were not controlled to match for experience or randomize across data types, and this may be a source of bias and/or variability in the comparison across manual processing sets. However, in our experience, this represents a common and naturalistic approach to performing large-scale data processing, and as such, the reported data may be considered as representative. Regarding failure modes for automated processing, we did not undertake a systematic analysis of what factors contribute to potential failure. For example, spatial or temporal resolution, in particular, resolution relative to body/aorta size, may influence the processing performance. In this study, a lower success rate for processing pediatric datasets than for processing non-pediatric datasets was observed. However, overall, the number of failures in this study was low and not conducive to conclusively identifying any such influences. Further investigation to evaluate the accuracy of automated or manual quantification relative to these factors, such as pediatric vs. adult and image resolution, would be valuable. The set of parameters we evaluated in this study are generally quantified on a voxel-wise basis, and they give only a limited reflection of potential flow structures such as vortices or helical flow that can exist in the heart and aorta. Future work to evaluate the accuracy of quantifying such parameters with this automated processing approach is required for any extension to analysis with complex flow dynamics.

## 5. Conclusions

Our primary findings include that the application of deep learning for automation, coupled with a modular processing pipeline framework, allows for a reliable automated quantification of aortic hemodynamics from 4D flow MRI data. This flexible tool for automating the full array of processing steps required for a quantitative analysis of 4D flow MRI data enables the construction of studies with cohort sizes in the hundreds to thousands of subjects. By minimizing the degree of user input at all stages of processing, the creation of large-cohort studies using 4D flow MRI data is made feasible and practicable without requiring extravagant computational resources. Lastly, the complete automation of processing for quantitative MRI is an important factor for improving reproducibility. While establishing a high degree of reproducibility overall is a longstanding challenge for MRI, with system imperfections and site-specific factors potentially contributing to variability in quantifications, removing manual inputs establishes a stable baseline for improving measurement reliability.

## Figures and Tables

**Figure 1 bioengineering-12-00807-f001:**
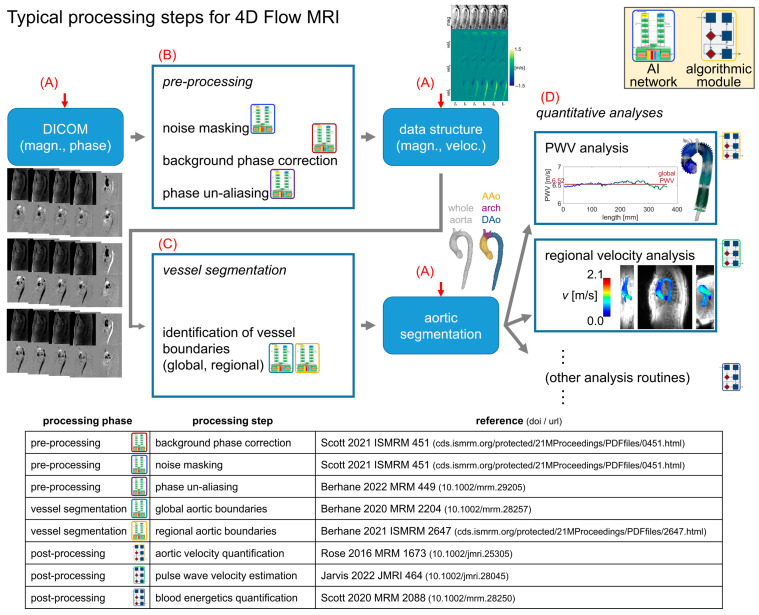
Standard processing flow. Preprocessing for 4D flow MRI follows a standard progression, with raw magnitude/phase data preprocessed before 3D segmentation of the aorta, followed by the quantification of hemodynamic parameters (e.g., PWV, *V*_max_). The data are stored in a standardized data structure, representing the image magnitude and velocity measurements. The processing pipeline framework automatically determines the appropriate entry point (red arrows) for a given dataset and then proceeds along the rest of the progression to quantify the hemodynamic parameters of interest with no user input (red letters A–D indicate tasks outlined in the Methods Section). Detailed references for the processing modules are shown in the table.

**Figure 2 bioengineering-12-00807-f002:**
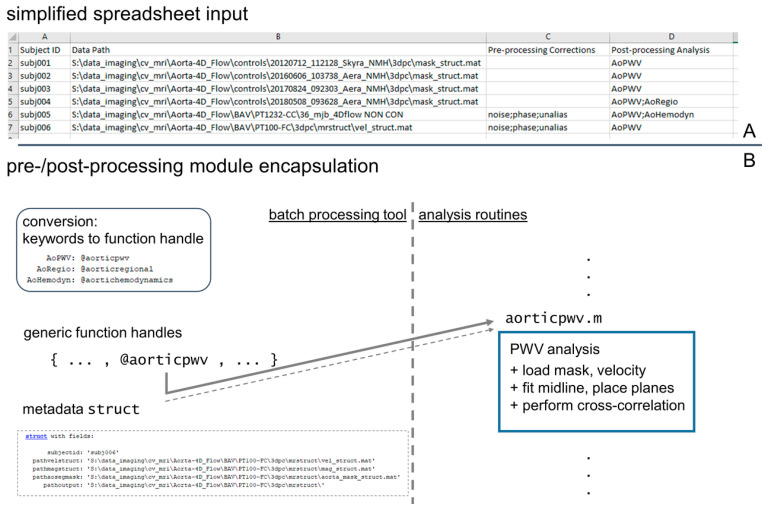
Input specification and module structure for processing tool. A simplified spreadsheet input (**A**) is used to specify datasets to be processed by the pipeline tool, and its creation constitutes the only user interaction needed. The spreadsheet contains columns identifying a dataset (“Subject ID” and “Data Path”) and specifying pre- and post-processing operations that are required for the dataset. For the execution of the processing steps, all pre- and post-processing modules are encapsulated with a uniform function header to allow for generic handling by the pipeline tool, which parses the spreadsheet and creates a queue of processing steps to complete (**B**).

**Figure 3 bioengineering-12-00807-f003:**
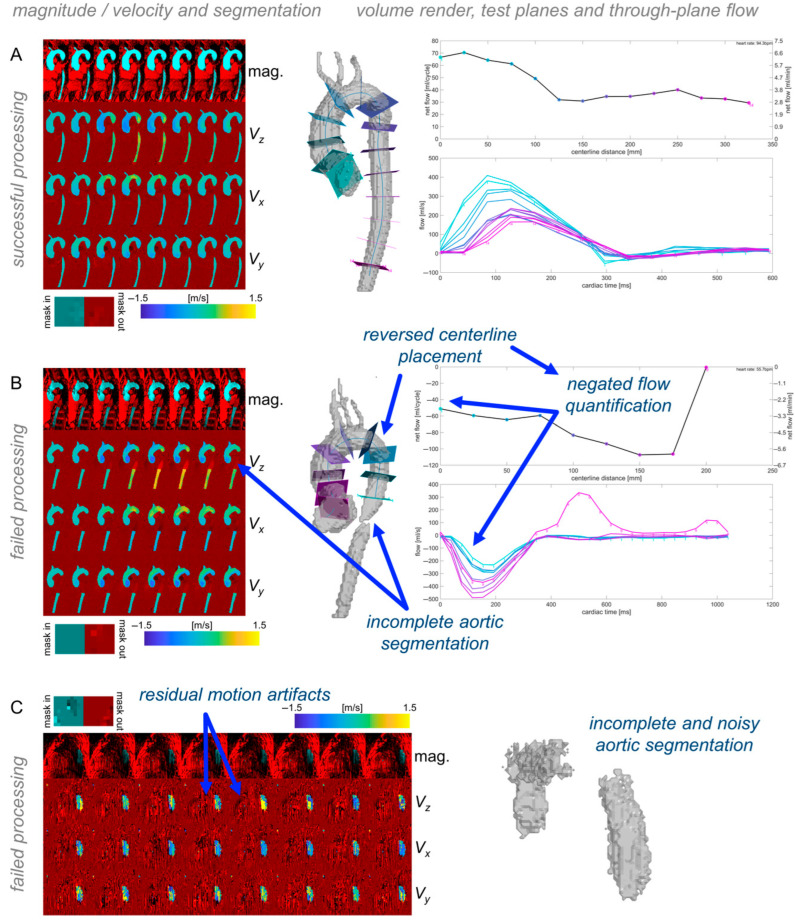
Quality check outputs. Each dataset’s quality check outputs show overlays of the aortic segmentation with magnitude and velocity images, and volume renderings of the segmentation with centerlines and through-plane flow along the length of the segmentation. Examples of successful processing (**A**) and different types of failed processing (**B**,**C**) illustrate how the quality check outputs can be used to quickly validate automated processing results.

**Figure 4 bioengineering-12-00807-f004:**
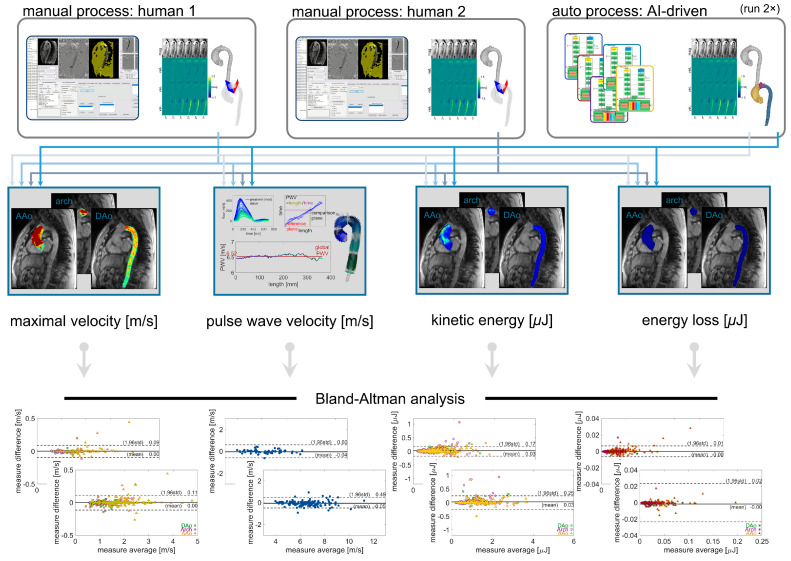
Comparison scheme for evaluating inter-observer reliability with manual or automated processing. Hemodynamic parameter quantifications for global aortic pulse wave velocity and regional aortic maximal velocity, kinetic energy, and energy loss, all derived from two sets of manual processing, were compared against each other and against the values derived from fully automated processing. The fully automated processing was performed twice to demonstrate the stability of quantification results.

**Figure 5 bioengineering-12-00807-f005:**
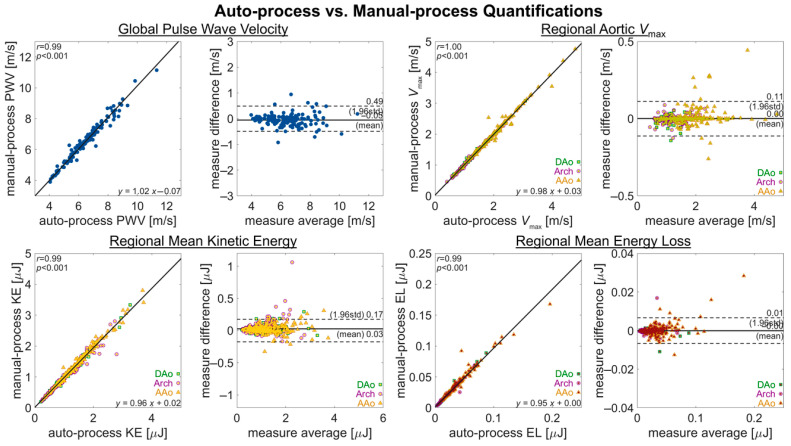
Measurement agreement for auto- and manual-processing quantifications. An assessment of measurement agreement, comparing quantifications from fully manual processing against those of fully automated processing for metrics of PWV, *V*_max_, KE, and EL. The quantification of PWV is a global aortic metric, with one value per subject. The quantifications of *V*_max_, KE, and EL are as regional metrics (AAo, arch, DAo), with three values per subject.

**Figure 6 bioengineering-12-00807-f006:**
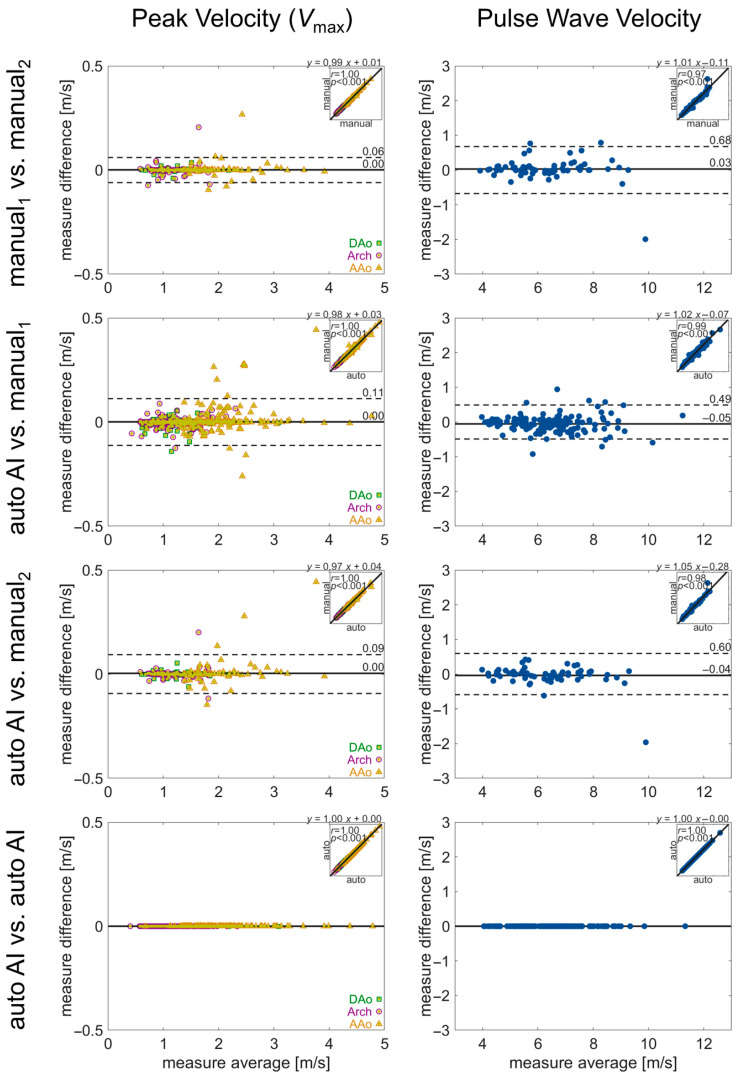
Evaluation of inter-observer reliability in manual and automated quantifications. Bland–Altman analysis for regional aortic volume-averaged pulse wave velocity and peak velocity. Solid lines show bias (mean difference) and dashed lines show limits of agreement (1.96 standard deviation difference). The insets in each plot show the correlation between the two quantifications, with the correlation coefficient and linear fit annotated. Values for the quantification in each aortic region (AAo, arch, DAo) are noted by a green square, purple circle, or yellow triangle, respectively.

**Figure 7 bioengineering-12-00807-f007:**
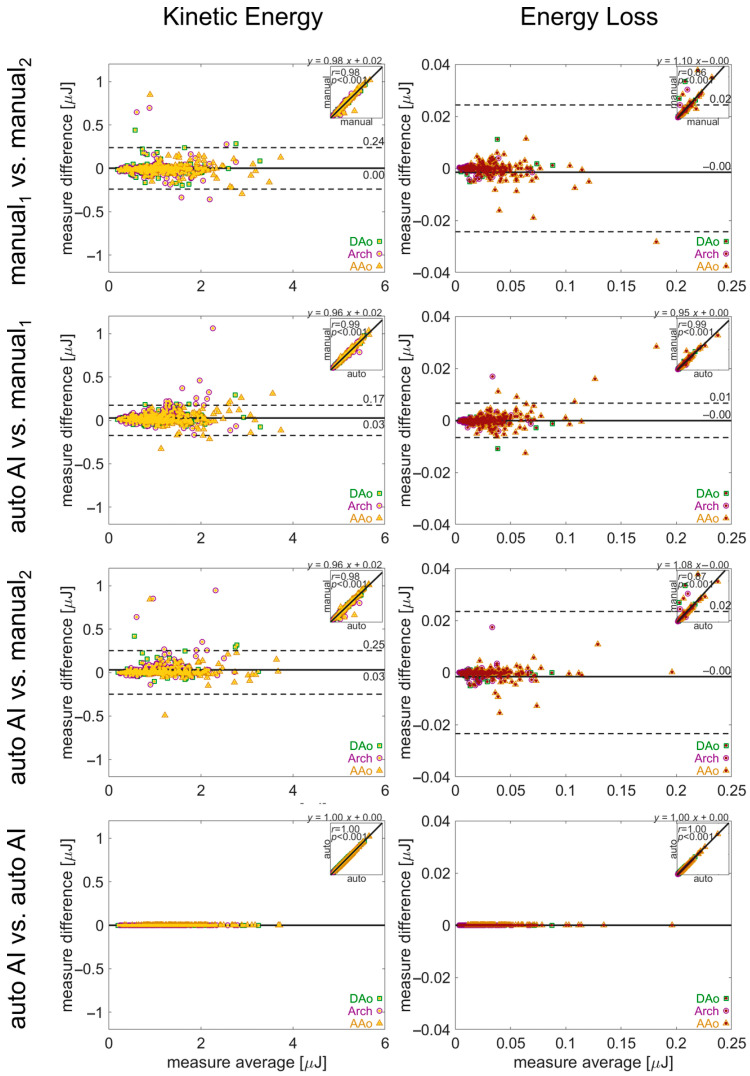
Evaluation of inter-observer reliability in manual and automated quantifications. Bland–Altman analysis for aortic kinetic energy and energy loss. The solid lines show bias (mean difference) and the dashed lines show limits of agreement (1.96 standard deviation difference). The insets in each plot show the correlation between the two quantifications, with the correlation coefficient and linear fit annotated. Values for the quantification in each aortic region (AAo, arch, DAo) are noted by a green square, purple circle, or yellow triangle, respectively.

**Figure 8 bioengineering-12-00807-f008:**
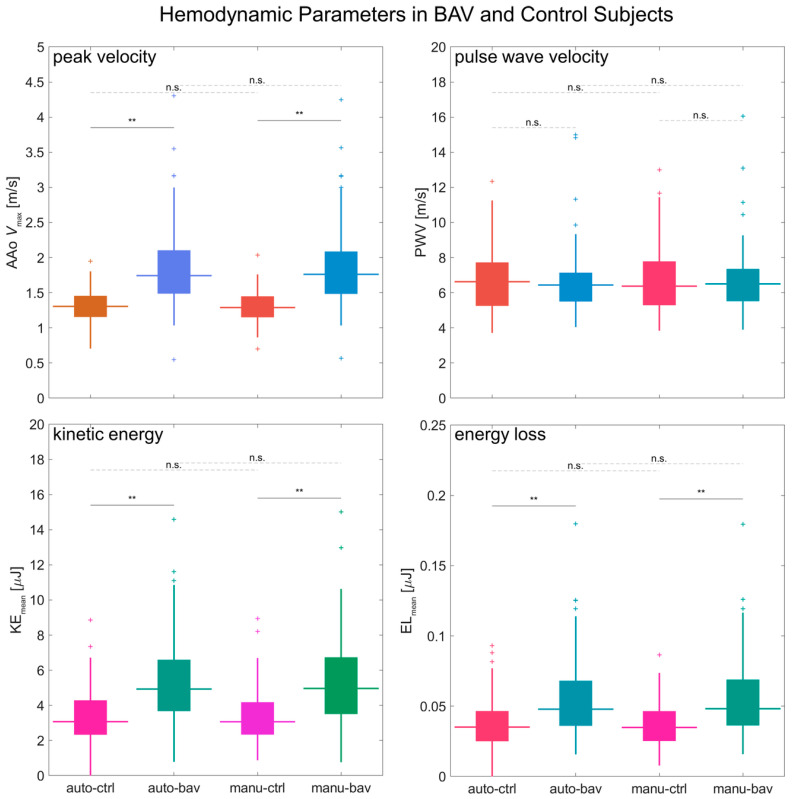
A comparison of hemodynamic parameters between BAV and controls. Boxplots for a group-wise (BAV vs. healthy) comparison of the *V*_max_, PWV, KE, and EL metrics calculated via automated and manual processing. Automated-processing values are labeled as “auto-” and manual-processing values as “manu-”; healthy subjects’ values are labeled as “ctrl” and BAV as “bav”. The horizontal lines above the boxplots show the *t*-test comparison results, with “n.s.” for *p* > 0.05 and “**” for *p* < 0.005 (no *p* values were observed in the range between 0.005 and 0.05).

**Table 1 bioengineering-12-00807-t001:** Summary of cohort and sub-cohort demographics. Subject counts and demographic distributions are shown for each subject type in the entire cohort (top), in which general processing success was evaluated, and in the sub-cohort (bottom), in which hemodynamic quantification reliability was assessed.

		Number of Subjects	Age (Mean ± Std.) [Years]	Sex (%) [Female]
**entire cohort**	**healthy control**	147	51.6 ± 16.9	60 (40.8%)
	**bicuspid aortic valve**	147	47.4 ± 12.5	32 (21.8%)
	**aortic valve prosthesis**	10	50.9 ± 11.3	2 (20.0%)
	**connective tissue disorder**	75	16.0 ± 4.3	28 (37.3%)
**sub-cohort**				
	**healthy control**	101	46.4 ± 15.5	51 (50.5%)
	**bicuspid aortic valve**	147	47.4 ± 12.5	32 (21.8%)

**Table 2 bioengineering-12-00807-t002:** Summary of automated processing success. A tally of all subject types in the evaluation cohort with datasets processed by the pipeline tool is given, with counts of datasets excluded due to segmentation errors noted for each subject type.

	Total	Included		Excluded	
healthy control	147	146	(99%)	1	(1%)
bicuspid aortic valve	147	143	(97%)	4	(3%)
aortic valve prosthesis	10	10	(100%)	0	(0%)
pediatric Marfan	62	55	(89%)	7	(11%)
pediatric Loeys-Dietz	11	9	(82%)	2	(18%)
pediatric Ehler-Danlos	2	2	(100%)	0	(0%)
*all groups*	*379*	*365*	*(96%)*	*14*	*(4%)*

## Data Availability

The data presented in this study are not publicly available due to privacy concerns but may be made available on reasonable request to the corresponding author.

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
