# Peer review of "A Fully Automated Analysis Pipeline for 4D Flow MRI in the Aorta"

_bioengineering, 2025, doi:10.3390/bioengineering12080807_

Round 1
Reviewer 1 Report
Comments and Suggestions for Authors
Fully-Automated Analysis Pipeline for 4d Flow MRI in the Aorta
Comments:
Abstract:
Introduction:
-
It is mentioned that the analysis manual takes between 10 and 20 minutes per dataset, even for expert users, without providing any references. It is recommended to support this claim with empirical data (your own or cited).
Methods:
-
It is stated that pre-trained networks are used, but it is not specified what data they were trained on or its availability.
-
The generation of quality figures (Figure 3) is mentioned, but it is not clear whether these were systematically reviewed by humans or how the quality of the segmentations was assessed.
Results:
-
It is reported that 14/379 studies were excluded due to "3D segmentation errors", but it does not explain what type of error it was or in what type of patients it occurred. Specify what caused the errors.
-
No data is reported on how long the pipeline takes to run or how long the manual analysis takes, despite this being one of the main promises of the article.
-
Although manual1 vs. manual2 is compared, it is not clear how many observers there were per group, whether they were randomly assigned, or their experience. Include a description of the observers, case distribution, and whether there were any bias control mechanisms (randomization, matching, pre-training).
-
It is stated that the pipeline gave consistent results between runs, but this repeatability is not quantified nor is the number of repetitions indicated.
Discussion:
-
4D cerebral flow analysis is mentioned as a potential example, but without bibliographic support or discussion of the anatomical or technical challenges of extrapolating the system to that region.
-
It is recommended to mention if there are any references that support the expected KE/EL values ​​or if a new approach is being proposed.
-
The lack of evaluation of execution time is not mentioned as a limitation (despite this being one of the arguments of the work).
Conclusions:
-
Although the pipeline is claimed to allow for studies with thousands of subjects, the study only included 379, and practical scalability (time, cost, computational resources) is not demonstrated.
-
The conclusion does not summarize the main results of the study. Include at least one final sentence mentioning the main findings and their clinical or methodological impact.
Reviewer 2 Report
Comments and Suggestions for Authors
The manuscript presents an AI-based approach to analysis of 4D flow MRI for the aorta. Agreement between parameters from the experimental analysis and those from the manual analysis is very good. The manuscript is well-written, and the presentation is good. I have the following comments for improvement.
- Please describe how long the manual analysis takes and how much time is save using the AI-based analysis.
- Please offer an explanation regarding why performance of the method is better in adult patients versus pediatric patients.
- Please describe how the kinetic energy, energy loss, and peak aortic velocities are used in clinical practice for the BAV and other patients evaluated in this report. Specifically, which treatment decisions depend on these parameters from 4D? How do the 4D-flow parameters measured add insights for treatment decisions relative to non-4D-flow measures from MRI and echocardiography in patients with a BAV? In what percentage of patients could the margin of error described with the automated method relative to the manual analysis lead to the wrong clinical treatment decision?
- Is the accuracy of the method relative to the manual method for adult and pediatric patients similar to the accuracy for control patients?
- Please describe automated methods for 4D flow available in commercial software. Have the authors performed a comparison of the method described in the paper with methods used in commercial software?
Reviewer 3 Report
Comments and Suggestions for Authors
The article presents a new tool that enables standardized processing of 4D Flow MRI data, completely eliminating the possibility of subjective expert assessments. The method was validated by comparing hemodynamics in the aorta with a normal aortic valve and a bicuspid aortic valve. The study is of significant practical interest; however, it does not account for the analysis of aortic flow considering the hydrodynamic structure of the blood flow. Emerging publications increasingly indicate that swirling flow in the heart and aorta significantly influences hemodynamics. Such flow is characterized by a number of quantitative parameters that should also be incorporated into the analysis algorithm.
Round 2
Reviewer 2 Report
Comments and Suggestions for Authors
The authors have responded appropriately to my comments and suggestions for revision.